# From Static to Dynamic: Adaptive Monte Carlo Search for Mathematical Process Supervision

## Abstract

The quality of process data plays a key role in training a Process Reward Model (PRM), which can enhance the complex mathematical reasoning capability of large language models. Existing methods estimate the quality of reasoning steps based on a fixed-budget sampling strategy and navigate a vast search space to perform path expansion during the automated data generation process, resulting in their inefficiency and inflexibility. To address these issues, we propose Adaptive Monte Carlo Search (AMCS), a framework that transforms data generation from fixed, static to adaptive, dynamic search at the level of node value estimation and path expansion. On one hand, AMCS adaptively refines estimation by allocating more samples to uncertain reasoning steps while using fewer samples for those that are easier to estimate. On the other hand, it enhances the path expansion through a Monte Carlo algorithm with a temporally adaptive policy that begins with broad exploration and gradually shifts toward exploiting the most promising directions. With AMCS, we construct a large-scale dataset `MathSearch-200K` of about 200K process supervision examples for training PRMs. To verify the effectiveness of our method, we conduct extensive experiments on four mathematical reasoning benchmarks. Experimental results show that Qwen2.5-Math-7B-PRM-AMCS achieves up to 76.2% accuracy on MATH500 with GLM-4-9B, outperforming all baseline PRMs. Notably, a 7B model supervised by Qwen2.5-Math-7B-PRM-AMCS surpasses a 72B model with weaker supervision. Moreover, Qwen2.5-Math-7B-PRM-AMCS maintains consistent advantages on out-of-distribution problems, demonstrating strong generalization capability. Our code is available at `https://github.com/reml-group/AMCS`.

## 1 Introduction

Large Language Models (LLMs) have demonstrated significant success across a wide range of natural language processing tasks (Ma et al., 2025a;b; Seo et al., 2025), including open-domain dialogue, summarization, and code generation. However, they often struggle with complex multi-step mathematical reasoning (Wang et al., 2024c), where precise logical consistency and error-free deduction are essential. This has motivated diverse efforts to improve reasoning capability, spanning architectural innovations (Zhan et al., 2025), targeted pre-training (Ren et al., 2025), post-hoc fine-tuning (Zhang et al., 2024), strategy prompting (Wu et al., 2024), and verification (Setlur et al., 2025a). Among these, verification is particularly appealing due to its model-agnostic nature and empirical effectiveness. By training a verifier to discriminate between correct and flawed reasoning paths, one can substantially enhance the LLM prediction, offering a scalable and generalizable avenue toward more trustworthy reasoning.

The verification in LLMs is broadly categorized into two paradigms: Outcome Reward Models (ORMs) and Process Reward Models (PRMs). ORMs (Cobbe et al., 2021b; Uesato et al., 2022) assign a scalar confidence score to an entire generated output, typically based on the final correctness or task success. In contrast, PRMs (Ma et al., 2023; Setlur et al., 2025b) evaluate the reasoning trajectory step by step, assigning intermediate rewards or correctness scores to each reasoning step. Recent studies (Yu et al., 2025; Ying et al., 2024; Wang et al., 2025) found that PRMs may outperform ORMs in the mathematical reasoning of LLMs due to the fine-grained step-level supervision

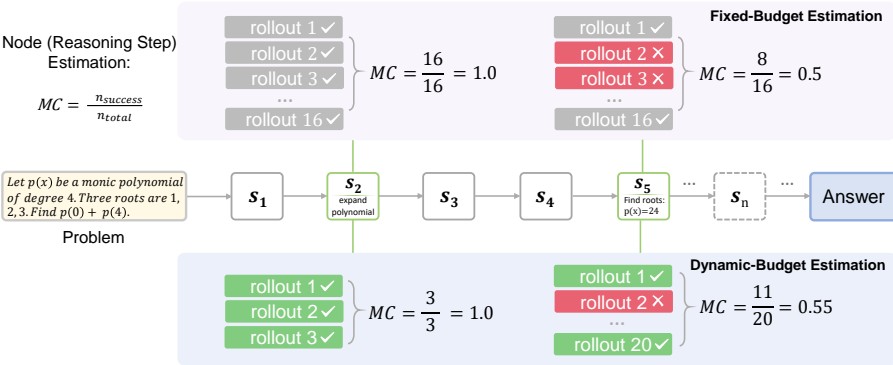

Figure 1: Strategy comparison of node (step) value estimation in automated process data generation. Fixed-budget approaches (top) allocate uniform resources across all nodes, while our dynamic-budget approach (bottom) adapts sampling effort based on node uncertainty.

and human-like cognitive evaluation. As we know, the primary bottleneck in scaling PRMs lies in data acquisition. High-quality process supervision requires value annotations for every reasoning step, which is an effort-intensive process that often demands substantial domain expertise, especially in complex, multi-step problems.

Although prior works (Wang et al., 2024a; Peng et al., 2025; Sun et al., 2025) leverage the Monte Carlo (MC) algorithm to obtain process labels automatically, they remain inefficient in node value estimation and inflexible in path expansion due to fixed sampling budgets and path expansion. For instance, as shown in Figure 1, they uniformly adopt a fixed budget of 16 samples per problem during node value estimation rather than dynamic sampling. In addition, they usually fail to balance exploration and exploitation during the path expansion, which is crucial for accurately localizing erroneous reasoning steps.

To address the mentioned issues, we propose Adaptive Monte Carlo Search (AMCS), a framework that transforms the data generation process from fixed, static to adaptive, dynamic search. Specifically, *to tackle the estimation inefficiency,* AMCS avoids allocating fixed computational effort to every reasoning node (step). Instead, it monitors evaluation uncertainty in real time and dynamically assigns more sampling resources to the nodes whose estimates remain uncertain. For nodes that have already converged to reliable estimates, our method reduces computational effort to improve overall efficiency. *To overcome the inflexibility of path expansion*, AMCS abandons the fixed exploration and exploitation strategy. It begins with broad exploration to uncover diverse reasoning paths. As the expansion progresses, it shifts toward exploiting the most promising directions, guided by accumulated evidence of path success. This adaptive expansion enables efficient generation of high-quality supervision data while reducing computational overhead. To verify the effectiveness and superiority of AMCS, we train a PRM model (Qwen2.5-Math-7B-PRM-AMCS) based on the generated large-scale process supervision data `MathSearch-200K` and conduct extensive experiments on AIME 2024/2025, MATH (Hendrycks et al., 2021), Olympiad-Bench (Li et al., 2024a), and Omni-MATH (Gao et al., 2024). Experimental results show that the combination of PRM-AMCS and GLM-4-9B consistently outperforms existing baselines, achieving 15.0%, 76.2%, 22.1%, and 19.0%, respectively.

Our main contributions are as follows:

- We propose an adaptive Monte Carlo search framework, which addresses the efficiency and inflexibility during the generation of process supervision data by introducing uncertainty-driven adaptive sampling and dynamic exploration and exploitation.

- We curate a high-quality process supervision dataset `MathSearch-200K` of 200K annotated reasoning trajectories with more precise value estimates and more reliable step-level supervision signals for challenging mathematical reasoning tasks.

- We conduct comprehensive experiments and analysis across four datasets, including reinforcement learning experiments, analysis of scaling, supervision, and adaptive allocation.

## 2 Preliminaries: Generation of Process Supervision Data

Fine-grained evaluation of intermediate reasoning steps is critical to train a high-quality PRM. Formally, given a dataset $\mathcal{D}$ consisting of large-scale tuples $(p, s_{1:t}, \hat{\mu}_t)$, the PRM is obtained by training on this dataset, where $p$ is a mathematical problem, $s_{1:t}$ is a partial reasoning trajectory up to step $t$, and $\hat{\mu}_t$ is a quality score reflecting the likelihood that the trajectory leads to a correct solution. Since obtaining $\hat{\mu}_t$ through expert annotation is prohibitively costly, prior works (Wang et al., 2024a; Peng et al., 2025; Sun et al., 2025) typically rely on automated MC-based pipelines to construct these supervisory signals.

The key idea is to evaluate the quality of any partial reasoning trajectory by measuring how often it leads to correct solutions. Specifically, given a partial reasoning sequence $s_{1:t}$ (representing the first $t$ steps of a solution attempt), the automated pipeline generates $N$ different expansions from this step and checks whether each expansion results in the correct final answer $a$. The quality score $\hat{\mu}_t$ is then estimated as the empirical success probability:

$$\hat{\mu}_t = \frac{1}{N} \sum \mathbb{I}(\text{expand}(s_{1:t})) = a). \tag{1}$$

As a concrete illustration, Appendix G presents a case study of rollouts on a math problem, showing the diversity of reasoning trajectories that arise from the same prompt. However, the mentioned pipeline reveals two limitations at different levels:

1. **At the Node Value Estimation Level.** Its reliance on a fixed-budget sampling strategy leads to inefficiency, as it ignores the varying difficulty of expansions from different nodes. For instance, in Figure 1, expanding from $\mathbf{s}_2$ is considerably easier than expanding from $\mathbf{s}_5$, suggesting that uniform exploration across all nodes is unnecessary.

2. **At the Path Expansion Level.** It overlooks the adaptive balance between exploration and exploitation in the search stage. This stage is critical for accurately localizing erroneous reasoning steps, which is essential for curating the dataset $\mathcal{D}$.

Therefore, the non-adaptive or fixed nature of both these levels is the efficiency and inflexibility bottleneck in current automated annotation pipelines.

## 3 Adaptive Monte Carlo Search

### 3.1 Overview

To overcome the aforementioned dual limitations, we introduce the Adaptive Monte Carlo Search (AMCS) framework, illustrated in Figure 2. At its core, AMCS reimagines the data generation process—shifting from a fixed, static paradigm to an adaptive, dynamic search strategy. The top panel of the figure depicts the dynamic estimation of node values based on the uncertainty, while the bottom panel illustrates the adaptive path expansion based on the trade-off between exploration and exploitation. After obtaining the process supervision dataset `MathSearch-200K`, we train a PRM model to be a verifier and employ it to guide LLMs to solve math problems.

### 3.2 Uncertainty-Driven Adaptive Sampling

To achieve a reliable estimate with the fewest samples, AMCS transforms the evaluation of a single node from a one-off, brute-force sampling into an adaptive iterative process that dynamically adjusts sampling effort based on estimation confidence.

**Initial Sampling with Rollout Clustering.** For any given reasoning prefix $s_{1:t}$, the adaptive process begins with a small, exploratory set of $k_{\text{init}}$ rollouts generated using non-greedy decoding. Under such stochastic generation, these initial rollouts often pursue different solutions (e.g., factorization vs. substitution for the same algebraic problem, or forward vs. backward reasoning for geometric proofs), making them fundamentally heterogeneous. As shown in Figure 2, treating such diverse rollouts as equivalent samples leads to inefficient estimation, since aggregating values from heterogeneous rollouts (e.g., $r_1, r_2$ vs. $r_3, r_k$) introduces additional variance that degrades value estimation accuracy.

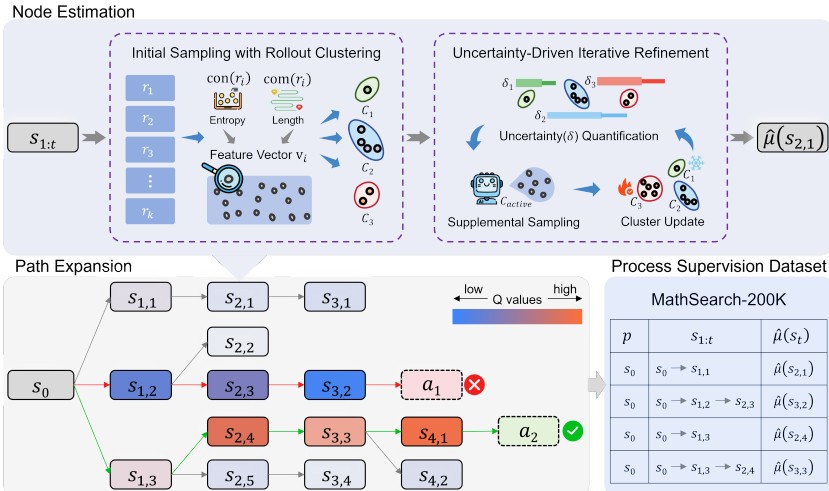

Figure 2: The AMCS framework. The top panel illustrates the adaptive process of node value estimation, which transitions from an initial exploratory sampling stage to an uncertainty-guided iterative refinement. The bottom panel shows the integration of this process within a step-wise path expansion process, where nodes are colored by their estimated Q-values.

Given this heterogeneity, a natural idea is to group rollouts following similar reasoning patterns together for more accurate success probability estimation within each group. Specifically, AMCS partitions the diverse initial rollouts into $K$ homogeneous clusters by featurizing each rollout $r_i$ with a two-dimensional feature vector $\mathbf{v}_i$ that characterizes both the generation confidence and the solution complexity:

$$\mathbf{v}_i = [\text{Confidence}(r_i), \text{Complexity}(r_i)], \tag{2}$$

where the generation confidence is measured by the average token-level negative log-likelihood $(-\frac{1}{L_r} \sum_{l=1}^{L_r} \log P(w_l^{(r)} | w_{<l}^{(r)}))$ to reflect the model's certainty during generation, and the solution complexity is captured by $\log(L_r + \zeta)$. Here, $L_r$ is the total number of tokens in the rollout, and $w_l^{(r)}$ denotes the $l$-th token in rollout $r_i$ and $\zeta = 10^{-6}$ prevents numerical issues. Since these features have different scales and units, z-score standardization is applied to ensure equal contribution to the distance-based clustering, as detailed in Appendix B.1. Based on the standardized feature representation, the K-Means algorithm is employed to partition the rollouts into $K$ strategy clusters $C = \{C_1, \dots, C_K\}$, where each cluster $C_j$ contains rollouts with similar confidence and complexity profiles, enabling more targeted estimation within homogeneous strategy groups.

**Uncertainty-Driven Iterative Refinement.** Building on the initial clustering results, our method iteratively refines success probability estimates through uncertainty-guided sampling. The core principle is that clusters with higher uncertainty require more samples to achieve reliable estimates, while confident clusters need minimal additional sampling. This ensures computational resources focus on clusters where additional samples provide the most information gain. The refinement process maintains success probability estimates $\hat{p}_j = s_j/n_j$ for each cluster $C_j$, where $s_j$ and $n_j$ denote successful and total rollouts, respectively, and quantifies estimation confidence through uncertainty measure $\delta_j$. Specifically, we employ the Wilson score interval to measure cluster-level uncertainty:

$$\delta_j = \frac{z}{1 + \frac{z^2}{n_j}} \sqrt{\frac{\hat{p}_j(1 - \hat{p}_j)}{n_j} + \frac{z^2}{4n_j^2}}, \tag{3}$$

where $z \approx 1.96$ for 95% confidence (Wilson, 1927). This formulation provides reliable confidence bounds even with small sample sizes or extreme probabilities (Brown et al., 2001). The node-level uncertainty $\delta_{\text{node}}$ aggregates cluster uncertainties weighted by sample sizes (detailed derivation in Appendix B).

At each iteration, we identify the target cluster $C^*$ with maximum uncertainty among active candidates. A cluster is considered active if it has not converged (uncertainty $\delta_j$ exceeds threshold $\epsilon_{\text{cluster}}$) and has remaining sampling budget (current samples $n_j$ below limit $n_{\text{max}}^{\text{cluster}}$). Formally, the active set is $\mathcal{C}_{\text{active}} = \{C_j : n_j < n_{\text{max}}^{\text{cluster}} \wedge \delta_j > \epsilon_{\text{cluster}}\}$, and the target cluster is selected as:

$$C^* = \underset{C_j \in \mathcal{C}_{\text{active}}}{\arg\max} \, \delta_j. \tag{4}$$

The number of new samples $m_{\text{step}}$ allocated to $C^*$ scales with its uncertainty level:

$$m_{\text{step}} = \min\{m_{\text{max}}, \max\{m_{\text{min}}, \lceil \gamma \cdot \delta_{C^*} \rceil\}\}, \tag{5}$$

where $\gamma$ converts uncertainty to sample counts, and bounds $[m_{\text{min}}, m_{\text{max}}]$ ensure reasonable batch sizes. After generating these rollouts and assigning them via feature distance, we update cluster statistics and proceed to the next iteration.

**Node Value Estimation.** The iterative refinement process from the above step does not continue indefinitely. To ensure computational efficiency and prevent excessive sampling on nodes that have already converged to reliable estimates, we establish a set of principled termination criteria. Specifically, the process terminates when any of the following three complementary conditions is met:

$$\delta_{\text{node}} \leq \epsilon_{\text{node}} \quad \text{or} \quad \sum_{j=1}^{K} n_j \geq k_{\text{max}} \quad \text{or} \quad \forall j : \delta_j \leq \epsilon_{\text{cluster}}, \tag{6}$$

each condition corresponding to confidence achievement, budget exhaustion, and universal cluster convergence, respectively. Upon termination, the final Monte Carlo estimate for node $s$ aggregates cluster-level success probability weighted by their respective sample sizes:

$$\hat{\mu}(s) = \sum_{j=1}^{K} \frac{n_j}{n_{\text{total}}} \cdot \hat{p}_j. \tag{7}$$

This weighted average ensures that clusters with more samples contribute proportionally more to the final estimate, reflecting their higher confidence levels. The value $\hat{\mu}(s)$ serves as the Monte Carlo value estimate for the node, which is subsequently used as the Q-value in the MCTS selection phase.

## 3.3 Adaptive Path Expansion

Building upon the adaptive node evaluation in Section 3.2, we employ adaptive path expansion to navigate the reasoning space.

**Selection Mechanism.** Following OmegaPRM (Luo et al., 2024), we select nodes to expand during tree search based on the exploitation value:

$$Q(s, r) = \alpha^{1-\hat{\mu}(s)} \cdot \beta^{\frac{\text{len}(r)}{L_p}}, \tag{8}$$

where $s$ represents a reasoning state and $r$ denotes a rollout continuation, $\hat{\mu}(s)$ is the node-level success estimate from Section 3.2, $\alpha, \beta \in (0, 1)$ are scaling factors, and $L_p$ is the normalized problem length. The exploration bonus employs the UCT principle (Kocsis & Szepesvári, 2006; Silver et al., 2016):

$$U(s, r) = c_{\text{puct}} \sqrt{\frac{\log N(s)}{1 + N(s, r)}}, \tag{9}$$

where $N(s, r)$ and $N(s)$ denote visit counts, and $c_{\text{puct}} > 0$ controls exploration strength.

**Dynamic Exploration-Exploitation Trade-off.** Beyond adaptive node estimation, we introduce temporal modulation of the exploration-exploitation balance. Traditional MCTS maintains fixed weighting throughout search, but for effective process supervision data generation, this balance should evolve as information accumulates. We define a time-varying selection score:

$$\pi_t(s, r) = (1 - w_t)Q(s, r) + w_t U(s, r), \quad w_t = \exp(-t/T), \tag{10}$$

where $t$ is the current iteration and $T > 0$ controls the transition rate. The exponentially decaying weight $w_t$ ensures exploration dominates initially when value estimates are uncertain, then progressively shifts to exploitation as confidence increases.

## 3.4 PROCESS REWARD MODEL TRAINING

Following the generation of process supervision data via AMCS, we train a process reward model designed to evaluate intermediate reasoning steps. The training dataset $\mathcal{D}$ comprises approximately 200,000 reasoning trajectories generated by applying AMCS to problems from MATH500 and GSM8K. Each training instance $(p, s_{1:t}, \hat{\mu}(s_t))$ consists of a problem $p$, a partial reasoning trajectory $s_{1:t}$, and its corresponding Monte Carlo value estimate $\hat{\mu}(s_t) \in [0, 1]$. We initialize the PRM with Qwen2.5-Math-7B-Instruct to leverage its strong mathematical reasoning capabilities. To fully leverage the continuous, fine-grained nature of the signals produced by AMCS, we employ a binary cross-entropy loss function with soft labels. In this formulation, the continuous Monte Carlo estimate $\hat{\mu}(s_t) \in [0, 1]$ directly serves as the target probability rather than being binarized. The training objective is:

$$\mathcal{L}(\theta) = -\frac{1}{|\mathcal{D}|} \sum_{(p, s_{1:t}, \hat{\mu}) \in \mathcal{D}} \left[ \hat{\mu} \log f_\theta(p, s_{1:t}) + (1 - \hat{\mu}) \log(1 - f_\theta(p, s_{1:t})) \right], \tag{11}$$

where $f_\theta(p, s_{1:t})$ represents the score predicted by the PRM. This soft label mechanism preserves the uncertainty information from our adaptive framework: high-confidence estimates (where $\hat{\mu}$ is near 0 or 1) provide strong supervision signals, while uncertain estimates (near 0.5) naturally contribute weaker gradients, effectively regularizing the training process.

## 4 EXPERIMENT

### 4.1 EXPERIMENTAL SETUP

We evaluate Qwen2.5-Math-7B-PRM-AMCS across diverse mathematical reasoning benchmarks including GSM8K (Cobbe et al., 2021a), MATH (Hendrycks et al., 2021), AIME, Olympiad-Bench (Li et al., 2024a), and OmniMATH (Gao et al., 2024). Our experiments test four actor models (GLM-4-9B, Phi-4-mini-Instruct, Llama-3.2-3B-Instruct, Qwen3-8B) with three search strategies (Beam Search, Best-of-N, MCTS) for inference evaluation, and conduct PPO fine-tuning using Qwen2.5-Math-7B-Instruct. We compare against open source PRMs, including Math-Shepherd, PRM800K, and Deepseek variants. AMCS parameters use $k_{\text{init}} = 6$, $k_{\text{max}} = 32$, $\epsilon = 0.1$, with $K = 3$ clusters. The experimental details are provided in Appendix C.

### 4.2 MAIN RESULTS

We evaluate the effectiveness of AMCS by training PRMs with our adaptive data generation framework and comparing their performance against existing PRMs across four mathematical reasoning benchmarks: AIME, MATH, Olympiad-Bench, and OmniMATH. In our experiments, PRMs guide four different actor models (GLM-4-9B, Phi-4-mini-Instruct, Llama-3.2-3B-Instruct, and Qwen3-8B) in generating reasoning trajectories. We test these models across three search strategies (Beam Search, Best-of-N, and MCTS) to demonstrate the generalizability of our approach. Table 1 presents comprehensive results across all model-strategy combinations.

PRMs trained with AMCS-generated data demonstrate consistent improvements across all experimental settings, with our method achieving peak performance of 76.2% on MATH, 15.0% on AIME,

Table 1: Mathematical reasoning performance of different PRMs across various search strategies. Models marked with † serve as the actor models, which are responsible for generating the reasoning trajectories. All results are reported in accuracy (%). Dataset names are abbreviated: MATH is MATH500, Oly. is Olympiad-Bench, Omni. denotes OmniMATH, and Avg. represents the average score.

| Strategy | Verifier | Llama-3.2-3B-Instruct† | | | | | Phi-4-mini-Instruct† | | | | |
|---|---|---|---|---|---|---|---|---|---|---|---|
| | | AIME | MATH | Oly. | Omni. | Avg. | AIME | MATH | Oly. | Omni. | Avg. |
| Beam Search | Qwen2.5-Math-7B-Instruct | 0.0 | 45.3 | 10.4 | 10.5 | 16.6 | 5.0 | 48.0 | 11.4 | 7.5 | 18.0 |
| | Llama3.1-8B-PRM-Deepseek | 6.7 | 39.0 | 7.1 | 8.0 | 15.2 | 1.7 | 43.6 | 11.1 | 8.0 | 16.1 |
| | Qwen2.5-Math-7B-PRM800K | 6.7 | 52.5 | 13.9 | 12.1 | 21.3 | 5.0 | 68.5 | 17.8 | 15.7 | 26.8 |
| | Math-Shepherd-Mistral-7B | 8.3 | 54.5 | 10.4 | 13.0 | 21.6 | 8.3 | 69.1 | 11.9 | 13.0 | 25.6 |
| | **Qwen2.5-Math-7B-PRM-AMCS** | **10.0** | **61.4** | **13.6** | **13.7** | **24.7** | **8.7** | 68.5 | **19.8** | **16.2** | **28.3** |
| Best-of-N | Qwen2.5-Math-7B-Instruct | 1.7 | 52.8 | 12.2 | 12.1 | 19.7 | 1.7 | 41.4 | 11.7 | 11.5 | 16.6 |
| | Llama3.1-8B-PRM-Deepseek | 3.3 | 49.0 | 10.4 | 10.5 | 18.3 | 3.3 | 51.2 | 11.7 | 11.6 | 19.5 |
| | Qwen2.5-Math-7B-PRM800K | 1.7 | 56.6 | 12.3 | 12.8 | 20.9 | 6.7 | 64.8 | 16.8 | 12.3 | 25.2 |
| | Math-Shepherd-Mistral-7B | 1.7 | 53.8 | 12.0 | 12.6 | 20.0 | 6.7 | 64.2 | 14.5 | 14.9 | 25.1 |
| | **Qwen2.5-Math-7B-PRM-AMCS** | **6.7** | **59.9** | **13.9** | **13.2** | **23.4** | 6.7 | **68.0** | **17.7** | **16.2** | **27.2** |
| MCTS | Qwen2.5-Math-7B-Instruct | 1.7 | 44.7 | 8.8 | 10.0 | 16.3 | 3.3 | 53.0 | 10.1 | 7.0 | 18.4 |
| | Llama3.1-8B-PRM-Deepseek | 5.0 | 40.0 | 7.6 | 9.0 | 15.4 | 3.3 | 43.2 | 10.5 | 8.2 | 16.3 |
| | Qwen2.5-Math-7B-PRM800K | 8.3 | 59.4 | 14.6 | 12.6 | 23.7 | 5.0 | 68.0 | 18.2 | 17.1 | 27.1 |
| | Math-Shepherd-Mistral-7B | 6.7 | 57.6 | 11.9 | 11.7 | 22.0 | 5.0 | 65.6 | 12.5 | 14.6 | 24.4 |
| | **Qwen2.5-Math-7B-PRM-AMCS** | **8.3** | **60.0** | **14.7** | 12.3 | **23.8** | **8.3** | **69.0** | **19.4** | **17.2** | **28.5** |

| Strategy | Verifier | Qwen3-8B† | | | | | GLM-4-9B† | | | | |
|---|---|---|---|---|---|---|---|---|---|---|---|
| | | AIME | MATH | Oly. | Omni. | Avg. | AIME | MATH | Oly. | Omni. | Avg. |
| Beam Search | Qwen2.5-Math-7B-Instruct | 1.7 | 42.0 | 9.1 | 11.0 | 16.0 | 8.3 | 71.1 | 18.0 | 17.0 | 28.6 |
| | Llama3.1-8B-PRM-Deepseek | 3.3 | 43.7 | 12.3 | 10.4 | 17.4 | 5.0 | 69.0 | 14.7 | 14.2 | 25.7 |
| | Qwen2.5-Math-7B-PRM800K | 3.3 | 42.0 | 11.6 | 11.2 | 17.0 | 13.3 | 75.4 | 19.4 | 12.3 | 30.1 |
| | Math-Shepherd-Mistral-7B | 0.0 | 47.2 | **13.5** | 13.9 | 18.7 | 6.7 | 73.0 | 19.6 | 14.8 | 28.5 |
| | **Qwen2.5-Math-7B-PRM-AMCS** | **6.7** | **49.4** | 13.2 | **14.1** | **20.9** | **13.3** | **76.0** | **20.3** | **19.2** | **32.2** |
| Best-of-N | Qwen2.5-Math-7B-Instruct | 0.0 | 42.4 | 12.0 | 13.7 | 17.0 | 6.7 | 75.0 | 19.0 | 16.2 | 29.2 |
| | Llama3.1-8B-PRM-Deepseek | 3.3 | 41.2 | 13.2 | 11.9 | 17.4 | 10.0 | 74.2 | 18.7 | 16.2 | 29.8 |
| | Qwen2.5-Math-7B-PRM800K | 1.7 | 41.8 | 13.2 | **14.2** | 17.7 | 11.7 | 76.2 | 18.2 | 17.6 | 30.9 |
| | Math-Shepherd-Mistral-7B | 3.3 | 45.0 | 14.0 | 8.0 | 17.6 | 8.3 | 77.2 | **19.6** | 15.5 | 30.2 |
| | **Qwen2.5-Math-7B-PRM-AMCS** | **5.0** | **47.8** | **17.2** | 13.9 | **21.0** | 11.7 | **77.8** | 19.3 | **17.8** | **31.7** |
| MCTS | Qwen2.5-Math-7B-Instruct | 3.3 | 46.5 | 9.2 | 12.3 | 17.8 | 6.7 | 70.4 | 16.7 | 15.5 | 27.3 |
| | Llama3.1-8B-PRM-Deepseek | 3.3 | 46.2 | 14.2 | 11.8 | 18.9 | 13.3 | 69.0 | 16.6 | 15.6 | 28.6 |
| | Qwen2.5-Math-7B-PRM800K | 1.7 | 42.6 | 11.9 | 14.4 | 17.7 | 13.3 | 76.0 | 21.2 | 18.7 | 32.3 |
| | Math-Shepherd-Mistral-7B | 5.0 | 50.6 | 14.6 | 13.6 | 21.0 | 3.3 | 74.2 | 19.5 | 16.0 | 28.3 |
| | **Qwen2.5-Math-7B-PRM-AMCS** | **6.7** | **51.2** | **14.9** | **14.7** | **21.9** | **15.0** | **76.2** | **22.1** | **19.0** | **33.1** |

22.1% on Olympiad-Bench, and 19.0% on OmniMATH using GLM-4-9B with MCTS. The improvements exhibit several notable patterns that provide insights into the effectiveness of adaptive data generation.

The benefits scale positively with model capacity, where larger models (GLM-4-9B, Qwen3-8B) consistently show more substantial improvements compared to smaller models (Phi-4-mini, Llama-3.2-3B). This suggests that AMCS-generated supervision data provides richer learning signals that larger models can better exploit. Additionally, the improvements are more pronounced on challenging benchmarks such as AIME and Olympiad-Bench, indicating that adaptive resource allocation during data generation particularly benefits complex multi-step reasoning scenarios where traditional fixed-budget approaches may under-sample critical reasoning paths.

Across different search strategies, AMCS maintains consistent advantages while revealing interesting interaction patterns. MCTS generally yields the highest absolute performance, but the relative improvements from AMCS remain substantial across Beam Search and Best-of-N as well, demonstrating that the quality gains are inherent to the supervision data rather than dependent on specific inference mechanisms.

## 4.3 PROCESS SUPERVISION VERSUS MODEL SCALE

To validate the generalizability of AMCS across different model capacities, we evaluate our approach using actor models ranging from 1.5B to 72B parameters. Figure 3 demonstrates that Qwen2.5-Math-7B-PRM-AMCS consistently outperforms all baseline methods across the entire parameter range on both MATH500 and GSM8K benchmarks. The performance advantages are particularly pronounced in smaller models, where Qwen2.5-Math-7B-PRM-AMCS achieves 53.4%

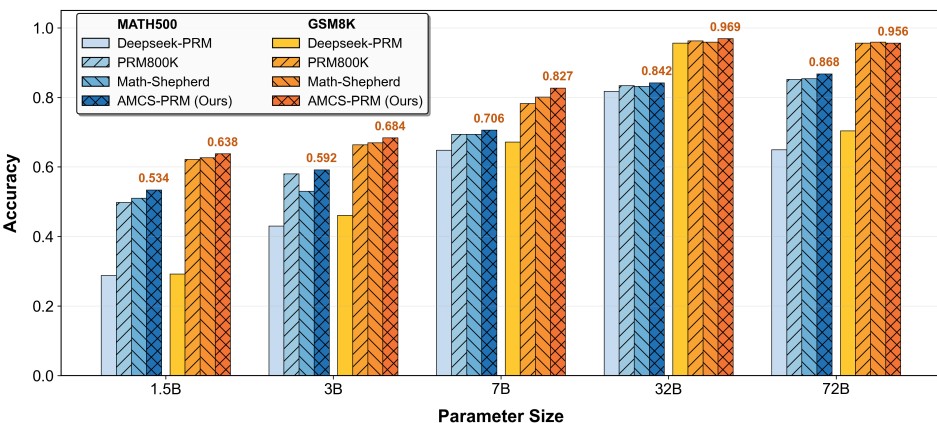

Figure 3: Performance comparison across Qwen actor models of different sizes (1.5B-72B) paired with various PRMs on MATH500 and GSM8K benchmarks.

accuracy on MATH500 with the 1.5B actor model compared to 28.8% for Deepseek-PRM, representing a 24.6 percentage point improvement. This substantial gap suggests that smaller models are especially sensitive to the quality of step-level supervision, making high-fidelity process rewards crucial for achieving competitive performance with limited parameters.

Remarkably, the scaling analysis reveals that superior process supervision can effectively compensate for reduced model capacity. A 7B model paired with Qwen2.5-Math-7B-PRM-AMCS (70.6% on MATH500) substantially outperforms a 72B model with weaker supervision (65.0% with Deepseek-PRM), despite requiring approximately $10\times$ fewer parameters. This finding indicates that investing in higher-quality process supervision may be more cost-effective than simply scaling model parameters. The consistent advantages maintained by AMCS across both math-specialized models (Qwen2.5-Math series) and general instruction-tuned variants (32B-Instruct) further demonstrate the robustness and broad applicability of our adaptive framework across different architectural choices and training paradigms.

### 4.4 SUPERVISION DATA ANALYSIS

Figure 4 examines the distribution characteristics of reasoning steps and token density across different process supervision datasets. AMCS exhibits a fundamentally different distribution profile compared to existing datasets, with a broader, right-skewed step distribution (mean: 11 steps) compared to the concentrated distributions of Math-Shepherd and PRM800K (6-7 steps). The token density analysis reveals systematic differences as well: AMCS averages 65 tokens per step with wider variance, indicating more detailed intermediate reasoning than baseline datasets (32-46 tokens). These distributional characteristics reflect important differences in data generation philosophy. The extended tail in AMCS step counts suggests systematic capture of complex reasoning scenarios that require multi-stage elaboration—cases potentially underrepresented in fixed-budget approaches. This adaptive granularity aligns with the intuition that mathematical problems exhibit varying intrinsic complexity, requiring correspondingly detailed supervision for effective process reward modeling.

### 4.5 ADAPTIVE ALLOCATION ANALYSIS

To understand the resource allocation behavior of AMCS, we analyze the sampling patterns across different node value ranges in our generated dataset. Figure 5 shows the distribution of MC rollouts and explored nodes across five value intervals. As illustrated in Figure 5(a), AMCS allocates significantly more rollouts to uncertain nodes ($\mu \in [0.4, 0.6]$: 20.0 rollouts) compared to confident ones ($\mu < 0.2$: 6.9 rollouts; $\mu > 0.8$: 7.1 rollouts), demonstrating a $3\times$ difference in sampling intensity. This adaptive allocation contrasts with the fixed 16-sample baseline (Luo et al., 2024; Wang et al., 2024a), which wastes resources on easy-to-evaluate extreme values while potentially undersampling uncertain regions. Similarly, as shown in Figure 5(b), the search depth varies from

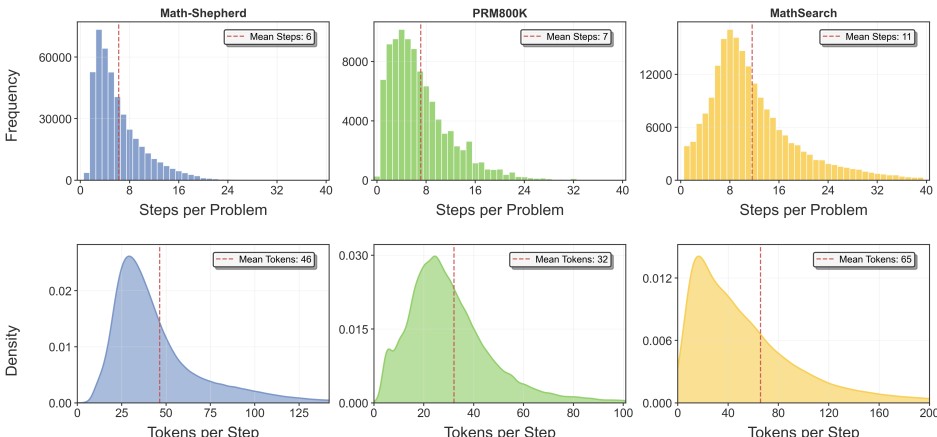

Figure 4: Distribution comparison of reasoning steps and token density across process supervision datasets. AMCS exhibits a fundamentally different distribution profile with a broader step distribution extending to longer reasoning sequences (mean: 11 steps) compared to the concentrated distributions of Math-Shepherd and PRM800K (6-7 steps). The token density analysis reveals that AMCS averages 65 tokens per step with wider variance, indicating more detailed intermediate reasoning than baseline datasets (32-46 tokens per step).

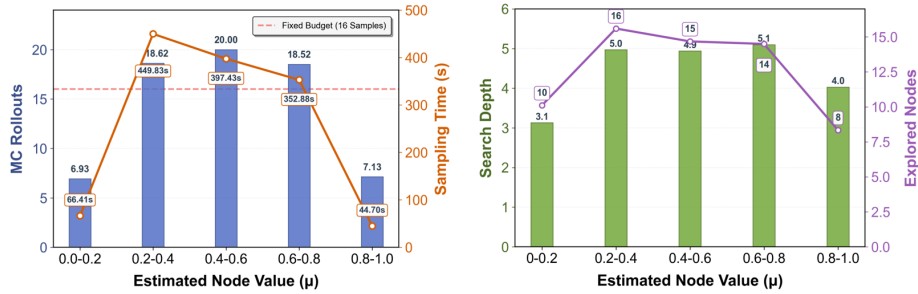

Figure 5: AMCS allocation patterns during data generation. (a) Distribution of MC rollouts per node across value ranges. (b) Search depth and total nodes explored for different node values.

3.1 for low-valued nodes to 5.0-5.1 for intermediate values, indicating that AMCS explores more extensively when facing higher uncertainty. The total nodes explored also peaks at intermediate values (14-16 nodes) versus extremes (8-10 nodes). We further provide qualitative analysis of reasoning steps across different value categories in Appendix F.

## 4.6 EFFICIENCY AND QUALITY ANALYSIS OF ADAPTIVE SAMPLING

To validate the efficiency and quality advantages of AMCS's adaptive sampling strategy, we sample 100 problems from MATH (Hendrycks et al., 2021), each with a 200-rollout budget, stratified by difficulty levels (Level 1=easiest, Level 5=hardest). Following OmegaPRM (Luo et al., 2024), we implement fixed-budget baselines with $k \in \{4, 10, 16, 22, 28\}$ using the OpenR framework (Wang et al., 2024b). Figure 6 presents computation time, search depth, and nodes explored.

The aggregate results (top row) show that AMCS maintains stable computational costs across problems (dashed lines), while fixed-budget strategies exhibit varying patterns. Notably, $k = 4$ incurs the longest time despite minimal per-node budget, as poor value estimates require exploring more nodes to find solutions. Larger $k$ values reduce explored nodes but increase per-node sampling, creating a breadth-precision trade-off. AMCS avoids this dilemma through uncertainty-driven allocation, balancing exploration and estimation under budget constraints.

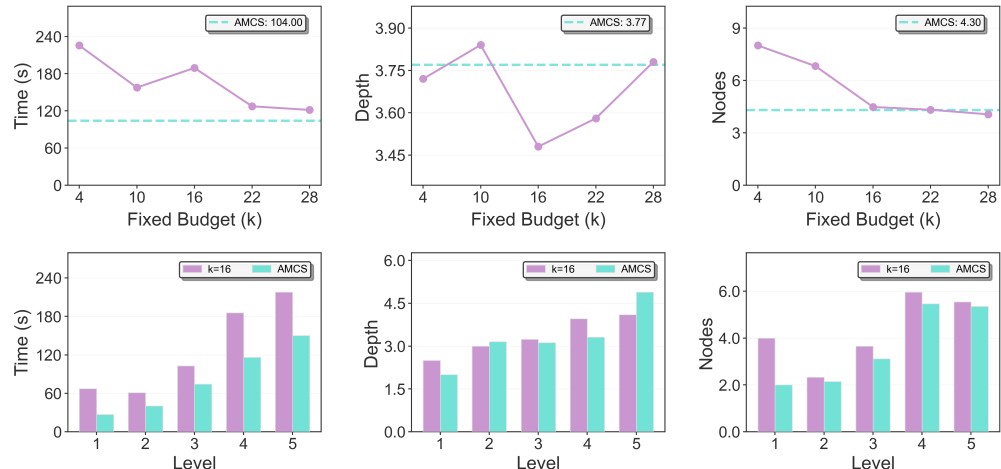

Figure 6: Computational efficiency comparison of AMCS against fixed-budget strategies following OmegaPRM (Luo et al., 2024). **Top**: Aggregate metrics showing AMCS's stable performance (dashed lines). **Bottom**: Comparison between $k = 16$ and AMCS across difficulty levels, demonstrating adaptive resource allocation.

The difficulty-stratified analysis (bottom row) reveals AMCS's adaptive behavior. Comparing $k = 16$ versus AMCS across levels shows that AMCS uses substantially less time and explores fewer nodes on simple problems (Levels 1-2), where quick convergence requires fewer samples. On hard problems (Levels 4-5), AMCS maintains comparable or greater depth, thoroughly exploring uncertain paths. This difficulty-aware allocation cannot be achieved with fixed budgets. Fixed budgets tend to waste resources on easy problems while leaving harder ones insufficiently sampled.

To assess the quality of value estimates, we analyze the distribution of estimated node values $\mu$ across the generated dataset using Kernel Density Estimation (KDE). Figure 7 compares three configurations: low fixed budget ($k = 10$), high fixed budget ($k = 28$), and AMCS. The setting with $k = 10$ exhibits a diffuse distribution with blurred peaks, resulting in many nodes assigned ambiguous intermediate scores ($\mu \approx 0.6$), which provide unclear supervision signals about step correctness. In contrast, both $k = 28$ and AMCS show concentrated distributions near extreme values, providing clear node evaluations. AMCS closely fits the distribution pattern of $k = 28$, achieving comparable estimation clarity

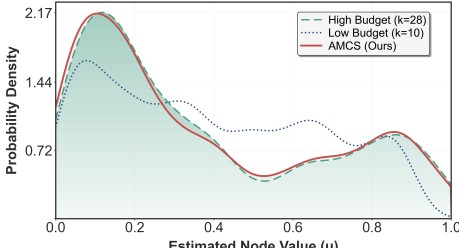

Figure 7: Node value distributions via KDE. AMCS closely aligns with the $k = 28$ setting showing concentrated peaks, contrasting with the $k = 10$ setting.

with fewer samples per node. This demonstrates that AMCS's adaptive allocation efficiently produces high-quality, high-confidence process supervision signals under budget constraints.

## 5 CONCLUSION

We propose an Adaptive Monte Carlo Search (AMCS) framework that reimagines the generation of process supervision data by shifting from fixed, static procedures to adaptive, dynamic search. On one hand, AMCS employs an uncertainty-driven adaptive sampling strategy to address the inefficiency inherent in node value estimation. On the other hand, it introduces adaptive path expansion to overcome the inflexibility of expansion. Leveraging AMCS, we curate `MathSearch-200K`, a dataset comprising 200K annotated reasoning trajectories, and utilize it to train a process reward model. Extensive experiments combining the reward model with large language models, using three distinct strategies across four benchmark datasets, demonstrate the effectiveness, superiority, and scalability of our approach.

## ETHICAL STATEMENT

This study is conducted in strict accordance with the principles of academic integrity. We affirm that all research presented is original and free from any form of plagiarism or data falsification. Throughout the research process, no personal or private information is involved. The objective of this work is to contribute positively to the advancement of the mathematical reasoning of large language models. We have thoroughly assessed the potential societal impact of our research and are confident that it poses no direct negative ethical risks. All authors have made substantial contributions to this study and have approved the final version of the manuscript for submission.

## REPRODUCIBLE STATEMENT

To ensure the reproducibility of this study, we provide all necessary code, data, and experimental configuration details. Code: All the implementation code, model scripts, and experimental procedures of this study have been open sourced on this website `https://anonymous.4open.science/r/AMCS-065C/`. The code repository includes detailed README.md files to guide environment configuration and code execution. Dataset: The core dataset used in this study is publicly available as an open-source resource and can be readily accessed for research purposes.

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

## A  RELATED WORK

**Mathematical Reasoning with LLMs.**  AI is advancing rapidly, with researchers pursuing human-like reasoning abilities in LLMs (Dasgupta et al., 2022). Mathematical reasoning serves as a key benchmark in this endeavor, requiring the integration of language understanding, symbolic manipulation, and multi-step reasoning with correct intermediate steps (Ahn et al., 2024). To address these challenges, prior work has explored several directions. 1) *Architectural innovations* introduce specialized components, such as subgoal decomposition or neuro-symbolic modules (Karpas et al., 2022; Li et al., 2024b), to bridge natural language understanding and formal mathematical computation. 2) *Targeted pre-training* on domain-specific or synthetic mathematical corpus (Wang et al., 2024d; Zhou et al., 2024) allows the model to learn structured reasoning patterns and symbolic manipulations that improve generalization on complex tasks (Lu et al., 2025; Shao et al., 2024). 3) *Post-hoc fine-tuning* further refines pretrained models with annotated reasoning traces, reflective feedback, or process-level supervision (Zelikman et al., 2022; Liu et al., 2023; Yan et al., 2025). 4) *Prompting strategies* guide models to generate intermediate steps or iteratively refine outputs without modifying model parameters, exemplified by chain-of-thought prompting (Wei et al., 2022), self-consistency (Wang et al., 2023), and rectification prompting (Wu et al., 2024). 5) *Verification methods* validate outputs through self-correction (Toh et al., 2023), external verifiers (Weng et al., 2023), or process-level evaluation (Liu et al., 2025), increasing the reliability and trustworthiness of model-generated solutions. While all these approaches improve reasoning, challenges such as error accumulation and unverified intermediate steps remain. This has drawn increasing attention to verification methods.

**Verification for Reasoning.**  Verification is crucial for improving the reliability of reasoning in LLMs, with two main paradigms: outcome reward models (ORMs) and process reward models (PRMs). ORMs assign rewards based solely on the correctness of the final answer and have been widely used in reinforcement learning with human feedback (RLHF) (Christiano et al., 2017). While effective for simple tasks, ORMs provide sparse feedback, which can reinforce spurious reasoning paths and limit performance in multi-step reasoning. By contrast, PRMs evaluate and reward intermediate reasoning steps, providing richer supervision that guides models toward correct reasoning trajectories. Empirical studies demonstrate the advantages of PRMs in various domains (Nath et al., 2025). In mathematics, WizardMath (Luo et al., 2025) and ThinkPRM (Khalifa et al., 2025) outperform ORM-based approaches on benchmarks including GSM8K (Cobbe et al., 2021b) and MATH-500 (Lightman et al., 2023), both in accuracy and data efficiency. In code generation, PRLCoder (Ye et al., 2025) and CODEPRM (Li et al., 2025), which incorporate execution feedback, achieve higher pass rates and better handling of complex tasks compared to ORM-guided reinforcement learning.

**Process Supervision Data Generation.** The effectiveness of PRMs depends on high-quality process supervision data. Traditional pipelines such as manual annotation, rule-based heuristics, or offline extraction provide supervision signals (Uesato et al., 2022; Lightman et al., 2023). Recent efforts have sought more scalable alternatives, for example, using Monte Carlo Tree Search (MCTS) to evaluate intermediate steps or leveraging verbalized verification chain-of-thought to reduce explicit labeling requirements (Wang et al., 2024a; Luo et al., 2024). While these approaches mitigate annotation costs, the generated supervision remains static and does not evolve with model behavior. In contrast, we propose a dynamic process supervision framework that continuously updates traces based on the model's evolving reasoning. This adaptive approach improves efficiency by focusing on uncertain or error-prone steps and enhances robustness under distribution shifts, overcoming the limitations of static supervision data.

# B  ALGORITHMIC DETAILS

This appendix provides an in-depth exposition of the key algorithmic components and implementation specifics of our Adaptive Monte Carlo Search (AMCS) framework. We detail the methodologies for quantifying uncertainty at both cluster and node levels, the process of feature engineering, and the robust assignment of new samples within the adaptive sampling loop.

## B.1  FEATURE ENGINEERING AND CLUSTER MANAGEMENT

This section details the feature extraction, standardization, and dynamic assignment procedures used in our adaptive Monte Carlo clustering framework.

**Feature Extraction and Standardization** For each rollout $r_i$, we extract a two-dimensional feature vector $\mathbf{v}_i = [\text{NLL}_i, \log(L_r + \zeta)]$ where:

- **Average Negative Log-Likelihood (NLL)**: $\text{NLL}_i = -\frac{1}{W_r} \sum_{j=1}^{W_r} \log P(w_j^{(r)} | w_{<j}^{(r)})$, where $W_r$ is the number of words in rollout $r_i$. This measures the model's generation confidence.
- **Log Complexity**: $\log(L_r + \zeta)$ where $L_r$ is the token length and $\zeta = 10^{-6}$ prevents numerical issues for very short rollouts.

Since these features operate on fundamentally different scales (NLL values typically range from 0.1 to 50+ while log-length ranges from 0 to 10), direct combination would result in NLL dominating the clustering distance calculations. To ensure both features contribute equally to the K-means clustering, we apply z-score standardization to the initial $k_0$ rollout features $\{\mathbf{v}_i\}_{i=1}^{k_0}$.

$$\hat{\mathbf{v}}_i = \frac{\mathbf{v}_i - \boldsymbol{\mu}_{\mathbf{v}}}{\boldsymbol{\sigma}_{\mathbf{v}} + \zeta_{\text{std}}} \tag{12}$$

where $\boldsymbol{\mu}_{\mathbf{v}} = \frac{1}{k_0} \sum_{i=1}^{k_0} \mathbf{v}_i$ and $\boldsymbol{\sigma}_{\mathbf{v}} = \sqrt{\frac{1}{k_0} \sum_{i=1}^{k_0} (\mathbf{v}_i - \boldsymbol{\mu}_{\mathbf{v}})^2}$ are computed element-wise. A small constant $\zeta_{\text{std}} = 10^{-8}$ is added to prevent division by zero for constant features.

The standardization parameters $(\boldsymbol{\mu}_{\mathbf{v}}, \boldsymbol{\sigma}_{\mathbf{v}})$ computed from the initial $k_0$ rollouts are stored and reused for standardizing features of subsequently generated rollouts during the adaptive refinement phase, ensuring consistent feature space representation throughout the clustering process.

**Dynamic Sample Assignment** During the iterative refinement phase, newly generated rollouts must be assigned to existing clusters. Each new rollout $r_{\text{new}}$ is assigned to the cluster whose centroid is closest in the standardized feature space:

$$\text{cluster}(r_{\text{new}}) = \underset{j \in \{1, \ldots, K\}}{\arg\min} \|\hat{\mathbf{v}}_{\text{new}} - \boldsymbol{\mu}_{C_j}\|_2 \tag{13}$$

where $\hat{\mathbf{v}}_{\text{new}}$ is the standardized feature vector of the new rollout and $\boldsymbol{\mu}_{C_j}$ denotes the centroid of cluster $C_j$ in the standardized feature space. The Euclidean distance (L2 norm) serves as the similarity metric.

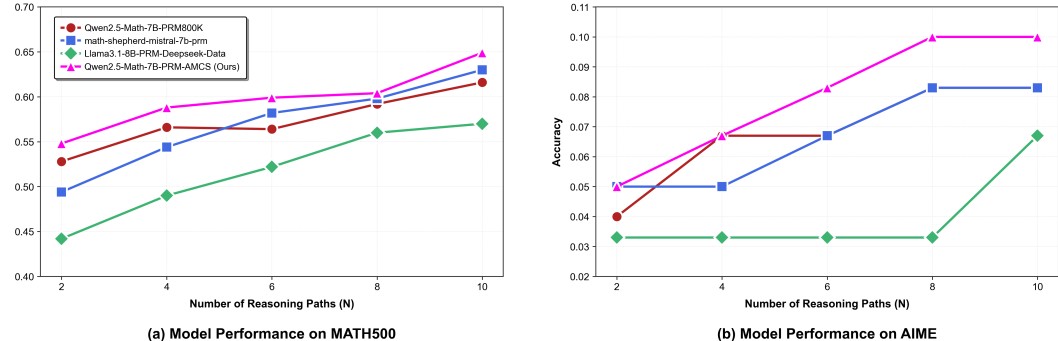

Figure 8: Performance comparison of four Process Reward Models (PRMs) on the (a) MATH500 and (b) AIME datasets. A unified actor model, Llama-3.2-3-Instruct, generates $N$ candidate reasoning paths. The final accuracy is determined by using each PRM for step-wise scoring to select the optimal path.

After assignment, the target cluster's statistics and centroid are updated incrementally:

1. The rollout index is added to the cluster's rollout list.

2. Success/failure statistics are recomputed based on all assigned rollouts.

3. The Wilson confidence interval and uncertainty measure are updated.

4. The centroid is recomputed as the mean of all standardized feature vectors assigned to the cluster.

This dynamic assignment mechanism ensures that newly generated rollouts are grouped with existing clusters representing similar reasoning strategies, maintaining the homogeneity principle essential for accurate uncertainty-driven sampling allocation.

### B.2 UNCERTAINTY QUANTIFICATION

In Section 3.2, we introduce the Wilson score interval for uncertainty quantification. Here we provide the complete mathematical derivation and additional technical details.

**Cluster-Level Uncertainty.** For each strategy cluster $C_j$, we compute a success probability estimate $\hat{p}_j = s_j/n_j$, where $s_j$ is the number of successful rollouts and $n_j$ is the total number of rollouts within cluster $C_j$. The Wilson score interval is derived from inverting the score test for a binomial proportion. For a binomial random variable with true probability $p$ and observed proportion $\hat{p}$, the score statistic is:

$$Z = \frac{\hat{p} - p}{\sqrt{p(1-p)/n}} \tag{14}$$

The Wilson confidence interval is obtained by solving $|Z| \leq z_{\alpha/2}$ for $p$, which yields the interval bounds. The uncertainty measure $\delta_j$ (Eq. 3 in the main text) represents half the width of this confidence interval. This formulation handles edge cases effectively: when $n_j$ is small, $\delta_j$ will be large, correctly indicating high uncertainty. Conversely, as $n_j$ increases, $\delta_j$ shrinks, reflecting increasing confidence in the estimate $\hat{p}_j$. The Wilson interval maintains valid coverage properties even with small sample sizes or probabilities near 0 or 1, which are common scenarios in our adaptive sampling setting where clusters may have few samples or exhibit very high/low success rates.

**Node-Level Uncertainty.** Beyond individual cluster uncertainties, we also require an overall uncertainty measure for the parent node $S_i$ that is currently being evaluated. This node-level uncertainty, denoted as $\delta_{\text{node}}$, aggregates the uncertainties from all active clusters within its scope, weighted by their relative contributions to the overall estimate. The overall node uncertainty $\delta_{\text{node}}$ is

Table 2: Performance comparison of the Qwen2.5-Math-7B-Instruct actor model when fine-tuned with PPO using different PRMs as the reward signal. All results are reported in accuracy (%). Here, pass@k denotes the proportion of problems for which a correct solution appears within the top-k generated outputs. For instance, pass@1 measures single-shot accuracy, while pass@5 allows up to five attempts.

| Reward Model | MATH500 | | GSM8K | | Hungarian Math | |
|---|---|---|---|---|---|---|
| | pass@1 | pass@5 | pass@1 | pass@5 | pass@1 | pass@5 |
| Qwen2.5-Math-PRM-7B | 55.6 | 72.6 | 80.0 | 93.4 | 46.9 | 65.6 |
| Qwen2.5-Math-7B-PRM800K | 53.4 | 72.8 | 82.2 | 97.4 | 43.8 | 62.5 |
| Skywork-o1-Open-PRM-Qwen-2.5-7B | 55.6 | 64.4 | 82.6 | 92.9 | 46.9 | 71.9 |
| **Qwen2.5-Math-7B-PRM-AMCS** | **61.6** | **73.2** | **83.5** | **97.5** | **53.1** | **75.0** |

computed as:

$$\delta_{\text{node}} = \sqrt{\sum_{j=1}^{K} \left(\frac{n_j}{n_{\text{total}}}\right)^2 \cdot \delta_j^2}. \tag{15}$$

This weighted combination reflects both the individual uncertainty inherent in each cluster's success probability estimate and the proportional influence of each cluster (based on its sample size $n_j$ relative to the total samples $n_{\text{total}}$) on the aggregated node value. A larger $\delta_{\text{node}}$ signifies higher overall uncertainty for the node $S_i$, indicating that its current Q-value estimate is less reliable and warrants further adaptive sampling to refine. This measure is critical for the confidence-based termination condition in Eq. 6.

## C EXPERIMENTAL DETAILS

**Datasets.** We evaluate on five benchmarks: GSM8K (Cobbe et al., 2021a) for grade school math, MATH (Hendrycks et al., 2021) for competition-level problems, AIME (60 problems from 2024-2025 American Invitational Mathematics Examination), Olympiad-Bench (Li et al., 2024a) for Olympic-difficulty problems, and OmniMATH (Gao et al., 2024) using 1/10 stratified sampling by difficulty.

**Model Configurations.** For inference evaluation, we test four actor models: GLM-4-9B (GLM et al., 2024), Phi-4-mini-Instruct, Llama-3.2-3B-Instruct, and Qwen3-8B (Team, 2025). PPO fine-tuning uses Qwen2.5-Math-7B-Instruct (Yang et al., 2024) as the base model. Scaling analysis covers the Qwen2.5 family from 1.5B to 72B parameters. We compare against multiple PRMs: for inference, we use Qwen2.5-Math-7B-Instruct, Llama3.1-8B-PRM-Deepseek-Data, Qwen2.5-Math-7B-PRM800K, and Math-Shepherd-Mistral-7B-PRM; for PPO training, we focus on Qwen-family PRMs, including Qwen2.5-Math-PRM-7B, Qwen2.5-Math-7B-PRM800K, and Skywork-o1-Open-PRM-Qwen-2.5-7B.

**Hyperparameters.** Inference uses three search strategies: Beam Search (beam size 5), Best-of-N (N=4), and MCTS (5 rollouts per node). PPO training employs a learning rate of 1e-6, batch size 4, and 3 epochs per update. AMCS parameters are set as: initial sampling $k_{\text{init}} = 6$, maximum budget $k_{\text{max}} = 32$, precision threshold $\epsilon = 0.1$, and $K = 3$ clusters. All experiments use consistent random seeds for reproducibility.

**Training Details.** During the data generation phase, four Tesla A800 GPU cards are used to train our data about one week. We use four Tesla A800 GPU cards to train a process reward model about three days.

**PRM Inference Strategies.** To clarify the inference procedures used in our experiments, we note that all inference methods operate at the step level rather than the token level. Each node in the search tree corresponds to a complete reasoning step generated by the LLM, and the PRM provides a scalar reward for that step; search decisions are therefore made over reasoning steps instead of individual tokens.

Table 3: Qualitative analysis of reasoning steps across different node value categories. Low-valued nodes ($\mu < 0.2$) typically contain straightforward calculations, while high-valued nodes ($\mu > 0.8$) often represent solution conclusions.

| Category | MC Value (μ) | Reasoning Step Content | Step Characteristics |
|---|---|---|---|
| MC↓ ROLLOUT↓ | 0.12 | Move all terms involving (2^x) to one side of the equation and constant terms to the other side. To do this, subtract (2^x) from both sides: | This type of step describes the specific **calculation or operation** to be performed next, followed by a demonstration of the operation, with a greater emphasis on calculation and execution. |
| | 0.12 | Each of these values will eventually reach 2. | |
| | 0.18 | Each time the center square is divided, the shaded area in the new smaller squares is a fraction of the area of the previous center square. | |
| MC↓ ROLLOUT↑ | 0.09 | To solve the equation, we start by analyzing the expression (x^2 + 2x + 3). We can rewrite it as: | These steps aim to pave the way for **subsequent complex calculations and reasoning**. This is a problem-solving plan that establishes the framework and direction for the entire problem-solving process. |
| | 0.18 | The given recurrence relation is (a{n+1} = a_n + \frac{a{n+2}}{2}). We can rearrange this to express (a{n+2}) in terms of (a_n) and (a{n+1}): | |
| | 0.18 | To find the greatest common factor (GCF) of (1001) and (2431), we can use the Euclidean algorithm. | |
| MC↑ ROLLOUT↓ | 0.90 | The only positive integer solution is ((m, n) = (1, 3)). | This type of step no longer involves any new reasoning or calculation, but rather declares **the results of the entire solving process**, which is a summary and affirmation of all previous work. |
| | 0.81 | Thus, the sum of all possible sums of the series is | |
| | 0.85 | Therefore, the number of different integer lengths for the third side is ( 9 ). | |
| MC↑ ROLLOUT↑ | 0.81 | However, since (a < d), this case is not valid. | This type of step is both **a summary of the previous part of the work** (such as eliminating an invalid situation) and **a preview of the next step of the work**. Provide direction for a complex, multi-stage problem-solving process. |
| | 0.81 | Step 3: Determine the shape of the regions formed. | |
| | 0.78 | Therefore, the foci of the hyperbola are at ((\pm 3, 0)).Step 2: Find the foci of the ellipse. The given ellipse is (\frac{x^2}{16} + \frac{y^2}{b^2} = 1). | |

*Beam Search.* We adopt step-level beam search following (Lightman et al., 2023) (Xie et al., 2023): at each expansion layer, the algorithm keeps the top-$k$ partial reasoning paths ranked by their PRM scores and discards the rest. This differs from vanilla token-level beam search, as pruning is performed at reasoning-step boundaries rather than at every token.

*Best-of-$N$.* We sample $N$ complete reasoning trajectories independently using the actor model. The PRM assigns step-wise rewards to each trajectory, and the trajectory with the highest final PRM score is selected. No intermediate pruning or tree search is performed.

*MCTS.* Our MCTS implements an AlphaZero-style step-level tree search (Silver et al., 2017): edges correspond to reasoning steps, PRM outputs serve as Q-value estimates for child nodes, and a UCB rule balances exploration and exploitation. This design enables selective exploration of promising reasoning branches under PRM guidance.

# D    REINFORCEMENT LEARNING WITH AMCS-TRAINED PRMS

To demonstrate the practical utility of AMCS beyond inference-time verification, we evaluate whether PRMs trained with our adaptive data generation framework can serve as more effective reward models in reinforcement learning settings. We conduct PPO fine-tuning experiments on Qwen2.5-Math-7B-Instruct, comparing our Qwen2.5-Math-7B-PRM-AMCS against three baseline PRMs from the same Qwen model family to ensure fair comparison: Qwen2.5-Math-PRM-7B, Qwen2.5-Math-7B-PRM800K, and Skywork-o1-Open-PRM-Qwen-2.5-7B. All experiments follow identical PPO training procedures with step-level reward supervision, varying only the reward model across conditions. Table 2 presents the performance comparison across different reward models on the MATH500, GSM8K, and the Hungarian Math out-of-distribution (OOD) benchmarks. Our approach achieves pass@1 (pass@5) scores of 61.6% (73.2%) on MATH500, 83.5% (97.5%) on GSM8K, and 53.1% (75.0%) on the Hungarian Math OOD dataset, consistently outperforming all baselines. The modest gain on GSM8K can be attributed to its less complex problems and the high baseline performance. In contrast, the substantial improvements on both the competition-level MATH500 and the OOD Hungarian Math are more significant. This demonstrates that the higher-quality process supervision provided by Qwen2.5-Math-7B-PRM-AMCS is especially beneficial for learning sophisticated and generalizable reasoning patterns, rather than just solving problems from a familiar distribution. These results provide crucial end-to-end validation, demonstrating that qual-

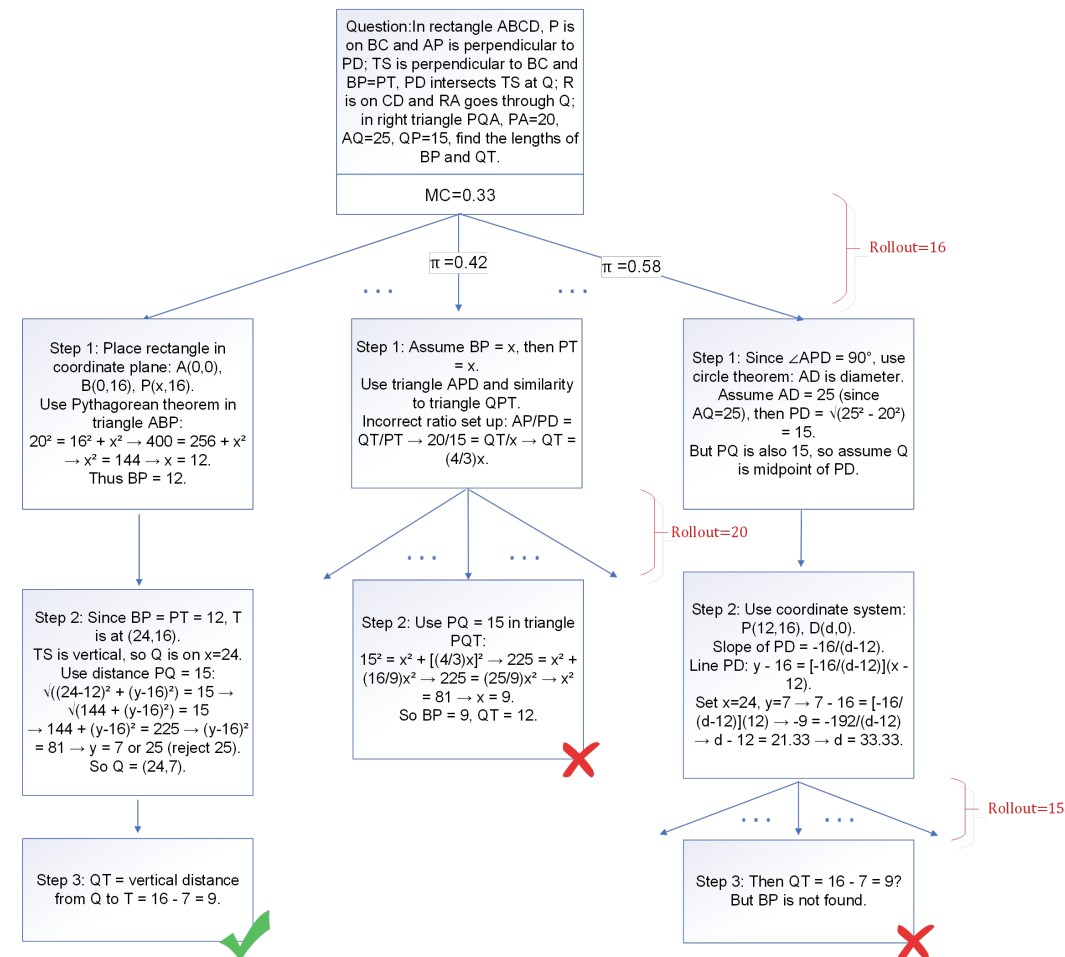

Figure 9: An illustrative rollout case showing multiple reasoning trajectories sampled from a single math problem. The figure highlights the diversity across rollouts, motivating the need for clustering and adaptive evaluation.

ity improvements in process supervision data directly translate into more capable and robust final models.

# E  EFFECT OF THE NUMBER OF REASONING PATHS

To evaluate the efficacy of our proposed Process Reward Model, Qwen2.5-Math-7B-PRM-AMCS, we benchmark its performance against three baseline PRMs: Qwen2.5-Math-7B-PRM800K, math-shepherd-mistral-7b-prm, and Llama3.1-8B-PRM-Deepseek-Data. We employ a unified actor model, Llama-3.2-3B-Instruct, to generate N candidate reasoning paths for each problem from two challenging mathematics competition datasets, MATH500 and AIME. The final accuracy is determined by using each PRM to perform step-wise scoring and select the best path from the candidate pool, with N varying from 2 to 10. The results, depicted in Figure 8, show a consistent trend where a larger N leads to higher final accuracy across all models. This aligns with the fundamental principle of Best-of-N sampling, where a larger candidate pool provides a higher performance ceiling. Crucially, our Qwen2.5-Math-7B-PRM-AMCS model consistently achieves the highest accuracy across all values of N on both datasets. This performance advantage is particularly pronounced on the more difficult AIME dataset, underscoring the robustness of our model. These findings demonstrate the superior discriminative capability of our proposed PRM, indicating that it provides more accurate step-wise reward signals for identifying high-quality reasoning processes compared to the baselines.

## F    REASONING STEP CHARACTERISTICS

To better understand the relationship between node values and reasoning complexity, we analyze the content characteristics of reasoning steps across different value categories. Table 3 presents representative examples from each category.

## G    CASE STUDY OF ROLLOUT DIVERSITY

To complement the discussion in Preliminaries, we provide a concise case study of reasoning rollouts sampled from a single math problem. As shown in Figure 9, several representative trajectories are depicted (with omissions for brevity), reflecting the inherent diversity of the rollout process.

