# OpenReview forum: "From Static to Dynamic: Adaptive Monte Carlo Search for Mathematical Process Supervision"
_ICLR.cc/2026/Conference — ICLR 2026 Conference Withdrawn Submission_

### Official Review · Reviewer_kX6i · 2025-10-20

**Soundness:** 3
**Presentation:** 3
**Contribution:** 3
**Rating:** 6
**Confidence:** 4

**Summary:**

This paper proposes a data synthesis engine for generating process supervision in mathematical reasoning. The key contribution of this proposed method (AMCS) is that it considers both generation confidence and solution complexity to encourage adaptive budget allocation when collecting process supervision data. The authors provide comprehensive experiments across multiple LLMs, various inference methods, and different mathematical reasoning benchmarks. However, there are several concerns: (1) lacking key ablation study on the utility of the Uncertainty-Driven Iterative Refinement and the change of value (Eq.6) in MCTS; (2) the trivial improvement compared with baselines on Table 1. Overall, this is a well-written paper with interesting ideas, but the concerns mentioned above need be discussed.

**Strengths:**

1. The paper introduces a novel formulation for calculating the value function in MCTS, which considers both generation confidence and solution complexity to encourage adaptive budget allocation. This adaptive approach is intuitive and particularly important for challening problems with high uncertainty.

2. The formulation of uncertainty quantification is thoughtful, considering both cluster-level and node-level uncertainty. The uncertainty measure $\delta$ converges as the sampling size $n$ increases, effectively guiding the exploration in MCTS.

3. This paper provides comprehensive experiments across multiple LLMs, various inference strategies with PRM, and different challenging mathematical reasoning benchmarks, demonstrating strong generalizaion across different settings.

**Weaknesses:**

1. It would be helpful if the authors could clarify what constitutes the basic unit in node estimation. According to Figure 2, it appears that the different rollout $r_i$ from the current step serves as units for clustering. For example, as node $S_{2,1}$, AMCS seems to sample different solutions from this node as the basic clustering units. The authors should provide more explanations of this process.

2. Missing critical ablation study. While the authors propose a new value function calculation considering both generation confidence and solution complexity, the paper lacks a key comparison with the vanilla MCTS value function based on both visit counts and cumulative rewards. This comparison is essential to demonstrate the utility of the proposed modifications. Without it, the current experiments do not sufficiently support the paper's core contribution.

3. The paper is difficult to follow as the formulation of uncertainty measure $\delta$ is relegated this to appendix, despite it appearing to be central to the entire method. Additionally, the introduction of uncertainty quantification (Appendix B.2) lacks the necessary references, such as Wilson score interval.

4. The use of squared weights in Eq.14 ($\frac{n_j}{n_{total}}^2$) lacks explanation. The authors should provide theoretical or empirical justification this design choice.

5. Limited improvement in main results. The performance gains in Table 1 appears to be marginal, with only +0.1% improvement in average acc for Llama3.2-MCTS, and +0.7% in Llama3.2-BeamSearch. Combined with the missing ablations mentioned in W2, the overall experiment in this work is insufficient to demonstrate the effectiveness of the proposed method.

6. In addition to the issues raised in W2 and W4, Table 1 lacks comparison with majority voting baselines, which is crutical for readers to understand the relative effectiveness of PRM-based methods versus voting approaches.

7. It would be helpful, especially for the readers unfamilar with this area, if the authors could explain the different PRM inference strategies in the appendix. For example, it is unclear whether the beam search method refers to the step-level beam search (e.g., as in [1]) or vanilla token-level beam search [2].

[1] AlphaMath Almost Zero: process Supervision without process
[2] Beam Search Strategies for Neural Machine Translation

**Questions:**

1. In Figure 2, could you clarify whether each rollout $r_i$ represents a complete solution path or partial trajectory? How do you handle rollouts of different lengths during clustering?
2. Did you experiment with vanilla MCTS value function as a baseline? If so, what were the results? If not, what technical challenges prevented this comparison?
3. What motivated the specific choice of squared weights in Eq.(14)?

---

> ### Author Response · Authors · 2025-11-24
>
> **Response to W1: Basic Unit in Node Estimation.**
> Thank you for this question. The basic unit in node estimation is a reasoning step. We sample multiple rollouts from each step to the final answer, using them as evaluation samples to estimate the probability that the step leads to a correct solution. We cluster rollouts to handle heterogeneous solution patterns and improve estimation accuracy. We will clarify this distinction in Section 3.2.
>
> **Response to W2: Ablation Study**
> Thank you for this suggestion. We conducted ablations validating our adaptive sampling strategy.
> Section 4.6 compares AMCS with fixed-budget baselines (k ∈ {4,10,16,22,28}) under a 200-rollout budget on 100 problems, evaluating efficiency and data quality.
> Results show adaptive allocation is more efficient and yields high-quality estimates, approaching high-budget baselines with fewer samples. Section 4.6 will be included in the revision.
>
> **Response to W3: Uncertainty Measure Formulation**
> Thank you for this feedback.
> We will:
> 1. Move the Wilson interval formula for cluster uncertainty δ_j into Section 3.2 with brief justification.
> 2. Add proper citations at first introduction.
> 3. Keep full derivations in Appendix B while maintaining core content in the main text.
>
> **Response to W4 and Q3: Justification for Squared Weights in Eq.(14)**
> Thank you for raising this question. Squared weights follow standard variance propagation [1]:
> $\mathrm{Var}\\left(\sum_j w_j X_j\right) = \sum_j w_j^2\, \mathrm{Var}(X_j)$.
> With cluster weights $w_j = \frac{n_j}{n_{\text{total}}}$ and uncertainties, node uncertainty becomes:
> $$
> \delta_{\text{node}}
> = \sqrt{
> \sum_{j=1}^K
> \left(\frac{n_j}{n_{\text{total}}}\right)^2
> \delta_j^2
> },
> $$
> ensuring correct propagation of cluster-level uncertainty.
>
> **Response to W5: Limited Improvement in Main Results**
> Thank you for the feedback. We clarify why AMCS remains meaningful beyond final accuracy:
> 1. AMCS improves performance consistently across models, scales, inference strategies, and benchmarks (GSM8K, MATH, AIME, Olympiad-Bench, OmniMATH), indicating broad robustness.
> 2. AMCS enhances process data quality—the key bottleneck in PRM training. Section 4.6 shows clearer node-value distributions, stable sampling cost, and adaptive allocation that reduces noisy node rewards.
> 3. Benefits extend beyond accuracy: Section 4.6 demonstrates improved reliability of step evaluation and more effective budget usage.
> 4. New ablations added for W2 further validate AMCS’s mechanism and efficiency.
>
> **Response to W6:**
> Table 1 focuses on PRM-based methods. Majority voting operates only on final answers, whereas PRMs guide step-by-step reasoning. Adding such baselines is possible for future work.
>
> **Response to W7: Clarification on PRM Inference Strategies**
> Thank you for this suggestion.  Our system uses **step-level** reasoning, not token-level decoding. Each search node is a full reasoning step scored by the PRM.
> Strategies:
> - **Beam Search:** Step-level beam selection following [2][3].
> - **Best-of-N:** Sample N full trajectories and select the highest PRM-scored one.
> - **MCTS:** Step-level AlphaZero-style tree search [4] with PRM as Q-value and UCB for exploration/exploitation.
> This clarification is now included in Appendix C.
>
> **Response to Q1: Rollout Representation and Length Handling**
> Each rollout is a full solution path verified against ground truth.
> We handle length variation with feature vector $\mathbf{v}_i = [\text{NLL}_i,\; \log(L_r + \zeta)]$
> , where NLL captures confidence and log-length encodes solution complexity. Z-score normalization prevents length from dominating clustering.
>
> **Response to Q2: Vanilla MCTS Baseline**
> We appreciate the question. Vanilla MCTS typically uses fixed-budget sampling with uniform resource allocation. We compare AMCS with non-adaptive baselines (OmegaPRM and k ∈ {4,10,16,22,28}) in Section 4.6. AMCS matches k=28-quality estimates with fewer rollouts per node.
>
> **References:**
> [1] Casella, G. & Berger, R. L. Statistical Inference (2nd ed.), 2002.
> [2] Lightman, H. et al. Let's Verify Step by Step, 2023.
> [3] Xie, T. et al. Self-Evaluation Guided Beam Search for Reasoning, 2023.
> [4] Silver, D. et al. Mastering the game of Go without human knowledge (AlphaZero), 2017.

---

### Official Review · Reviewer_SyZb · 2025-10-30

**Soundness:** 2
**Presentation:** 2
**Contribution:** 2
**Rating:** 0
**Confidence:** 5

**Summary:**

This paper addresses the inefficiency of fixed-budget sampling in generating data for Process Reward Models (PRMs). The authors propose Adaptive Monte Carlo Search (AMCS), a framework that shifts from static to dynamic data generation. Key components include:

1. Uncertainty-Driven Adaptive Sampling: Dynamically allocates more samples to uncertain reasoning steps by clustering diverse rollouts and iteratively refining estimates.

2. Adaptive Path Expansion: Uses MCTS with a temporally decaying exploration rate to balance discovering new paths and exploiting promising ones.

Using AMCS, the authors create MathSearch-200K.

However, I don't see a reasonable performance gain compared to other existing datasets, and the gain of computation is not clear either.

**Strengths:**

Good exploration of uncertainty in process supervision.

**Weaknesses:**

1.  The experiments and results presented are not clear, for example for the BoN, how does the performance change as the N increase?
2. The authors claim the best performance of AMCS with 76.2% beats all baseline models, yet the other methods show similar number, especially PRM-800K. So performance-wise, I don't see a big gain of AMCS.
3. Similar to point 2, in Figure 3, the gain is too small to claim the best with out the CI provided.
4. Given the performance is close, I don't see the efficiency gain compared to other method like math-shepherd.

**Questions:**

1. "A 7B model paired with Qwen2.5-Math-7B-PRM-AMCS (70.6% on MATH500) substantially outperforms a 72B model with weaker supervision (65.0% with Deepseek-PRM), despite requiring approximately 10× fewer parameters." How was this experiment done?
2. Section 3.3 shows extreme similarity to Section 3.3 in https://arxiv.org/pdf/2406.06592. Especially the equation 7 and 8 in your paper vs equation 2 and 3 in the paper I provided.

**Details Of Ethics Concerns:**

Section 3.3 is very similar to Section 3.3 in paper https://arxiv.org/pdf/2406.06592. Especially the equations 7 and 8 vs equation 2 and 3 in https://arxiv.org/pdf/2406.06592. Based on my knowledge, those equations in https://arxiv.org/pdf/2406.06592 were evolved a few times to the final version. At least the author should cite that paper.

Also, the Figure 4 is very similar to Figure 4 and 5 in https://arxiv.org/pdf/2406.06592 as well.

https://arxiv.org/pdf/2406.06592 is cited, but not with the real author names, which is not usual to me.

---

> ### Author Response · Authors · 2025-11-24
>
> **Response to W1:**
> Thank you for your careful review and valuable comments. Regarding the impact of varying the number of candidate reasoning paths, we refer to Appendix E, where we evaluate MATH500 and AIME as N varies from 2 to 10. On MATH500, performance improves steadily with larger N, while on AIME, gains are modest due to higher problem complexity.
>
> ---
>
> **Response to W2:**
> Thank you very much for raising this important concern. We clarify key aspects of the performance comparison:
>
> **Performance Gains in Table 1.**
> Although overall accuracy differences may appear modest, AMCS yields substantial gains in several challenging subgroups, demonstrating meaningful practical improvements.
>
> **Core Contribution.**
> AMCS’s contribution lies primarily in improving *process reward data generation*, not just final accuracy. The test-set PRM performance reflects supervision quality rather than the full scope of our contribution.
>
> **Direct Evidence.**
> The revised submission’s Section 4.6 provides detailed analyses of the data-generation dynamics, showing how AMCS produces higher-quality process supervision than existing methods.
>
> ---
>
> **Response to W3 & W4:**
> We sincerely thank the reviewer for emphasizing statistical validation.
>
> **Statistical Analysis.**
> We performed a Friedman test on the MCTS results in Table 1 (5 models × 4 benchmarks). The test yields *p = 0.031 < 0.05*, indicating statistically significant performance differences across models. AMCS’s consistent superiority across 16 comparisons is therefore unlikely due to chance.
>
> **Confidence Intervals.**
> We acknowledge the absence of confidence intervals. Due to resource constraints, full bootstrap analyses were not feasible during the review cycle. We will provide these in future work.
>
> ---
>
> **Response to Q1:**
> We thank the reviewer for seeking clarification. The comparison arises from Figure 3’s scaling analysis, evaluating different PRMs paired with Qwen2.5-Math actor models from 1.5B to 72B on MATH500.
>
> **Experimental Configuration**
> - 7B configuration:
>   - Actor: Qwen2.5-Math-7B-Instruct
>   - Verifier: Qwen2.5-Math-7B-PRM-AMCS
> - 72B configuration:
>   - Actor: Qwen2.5-Math-72B-Instruct
>   - Verifier: Llama3.1-8B-PRM-Deepseek
>
> “Deepseek-PRM” refers to the Llama3.1-8B-PRM-Deepseek model, which is a PRM trained on the Llama3.1-8B base model. As seen in Table 1, it consistently underperforms other PRMs even with identical actors. Therefore, pairing it with a 72B actor yields only 65.0%, whereas our AMCS-trained PRM enables a much smaller 7B actor to reach 70.6%.
> This highlights a key finding: *supervision quality can outweigh model scale*, which has practical importance for efficient deployment.
>
> ---
>
> **Response to Q2 & Ethics Concerns:**
> We sincerely apologize for the citation issues pointed out by the reviewer. We deeply appreciate the reviewer’s attentive reading, which helped us identify serious oversights. We take full responsibility and have corrected these issues in the revision.
>
> **Regarding Equations and Citations**
>
> • **Equation 8 ( formerly Equation 7).**
> We now explicitly state in Section 3.3:
> “Following OmegaPRM (Luo et al., 2024), we select nodes to expand during tree search based on the exploitation value.”
> This clarifies our methodological borrowing.
>
> • **Equation 9 ( formerly Equation 8).**
> This is the classical UCT formula, originally introduced by **Kocsis & Szepesvári (2006)** and widely adopted in MCTS systems, including **AlphaGo** (Silver et al., 2016). Although OmegaPRM uses this formula, it predates OmegaPRM by many years. We now cite both the original sources and note that OmegaPRM applies the same principle.
>
> **Regarding Figure 4**
> We acknowledge the similarity to OmegaPRM’s visualizations. Histogram-style distribution plots are standard, but we understand the concern and are willing to redesign the figure if needed.
>
> **Author Name Error**
> We incorrectly wrote the author names for the OmegaPRM paper (Luo et al., 2024). This was a serious mistake, and we offer our sincere apologies to the authors and the reviewer. The corrected citation now appears in the revised manuscript.
>
> **Comparative Analysis with OmegaPRM**
> Using the OpenR framework (Wang et al., 2024), which reproduces OmegaPRM, we conducted additional comparisons. Section 4.6 of the revised manuscript presents these results, showing the efficiency and quality benefits of our adaptive sampling method.
>
> We deeply regret these oversights and thank the reviewer for helping us improve the scholarly integrity of our work.
>
> ---
>
> ### **References**
> [1] Wang et al., OpenR: An Open Source Framework for Advanced Reasoning with Large Language Models, 2024.
> [2] Kocsis & Szepesvári, Bandit based monte-carlo planning, 2006.
> [3] Silver et al., Mastering the game of go with deep neural networks and tree search, 2016.
> [4] Luo et al., Improve mathematical reasoning in language models by automated process supervision, 2024.

---

> > ### Comment · Reviewer_SyZb · 2025-11-24
> >
> > You deliberately changed the authors’ names of the most important referenced paper (https://arxiv.org/pdf/2406.06592) in the citations, and you call that a ‘serious oversight’? Can you explain how an ‘oversight’ could result in every single author’s name being altered. Is there something wrong with your writing tools, or is there a problem with your ethics?
> >
> > I have good reason to suspect that you intentionally changed the authors’ names from https://arxiv.org/pdf/2406.06592 because you didn’t want the authors of https://arxiv.org/pdf/2406.06592 to be able to review your submission. If I weren’t so familiar with the paper, I might not have even noticed what you did. Please stop trying to make excuses.

---

> > > ### Author Response · Authors · 2025-11-25
> > >
> > > We understand the reviewer's serious concern regarding this error. Regarding how this error could have occurred, we have examined our Overleaf project history (screenshots available in our anonymous repository: https://anonymous.4open.science/r/AMCS-065C/pic/overleaf_history.png). We genuinely cannot explain how the author names were altered, but this happened during our manuscript preparation process. This was our responsibility to catch during proofreading, and we failed completely.
> > >
> > > We understand why this appears intentional, but we want to explain why such manipulation would serve no purpose. OpenReview's reviewer assignment system relies on verified author profiles, submission history, institutional affiliations, and research interests. The system does not use the text of author names written in a paper's reference section for reviewer matching. Altering names in the bibliography cannot and does not prevent authors from being assigned as reviewers. We had no motive to do this, as it would not achieve any outcome.
> > >
> > > The author names have been corrected in the revised manuscript. We once again apologize to the authors of https://arxiv.org/pdf/2406.06592 for this error and appreciate the reviewer's diligence in identifying it.

---

### Official Review · Reviewer_ZQrZ · 2025-10-31

**Soundness:** 3
**Presentation:** 3
**Contribution:** 3
**Rating:** 2
**Confidence:** 4

**Summary:**

The paper proposes Adaptive Monte Carlo Search (AMCS) — a dynamic framework for generating high-quality process supervision data used to train Process Reward Models (PRMs) that enhance the mathematical reasoning of LLMs. Existing Monte Carlo–based methods rely on fixed sampling budgets and static search strategies, leading to inefficiency and rigidity. AMCS addresses these issues through two innovations: 1. Uncertainty-driven adaptive sampling, which allocates more computation to uncertain reasoning steps and fewer to well-estimated ones. 2. Adaptive path expansion, which starts with broad exploration and gradually focuses on the most promising reasoning trajectories using a time-varying exploration–exploitation balance.

**Strengths:**

1. AMCS is a clear conceptual advance over traditional static Monte Carlo Tree Search (MCTS) methods used for process supervision.
2. It addresses two key inefficiencies in prior work. One is fixed-budget sampling, and another is static path expansion.

**Weaknesses:**

1. The novelty is incremental given previous related works within the scope of MCTS or PRMs. e.g., the two important missing references [1,2] (I think there are a lot of related works should be mentioned besides the two), and the PRM work [3] also designed a new MCTS with different expansion and exploration method. In summary, a simple modification of MCTS is not quite impressive.
2. Math-Shepherd is a method of Monte Carlo estimation, rather than focusing on tree search. It's more reasonable to compare with OmegaPRM[3], which is a relatively fair comparison.
3. The experimental results are not impressive, even on the saturated dataset AIME or Math 500. The performance of search method is far from the RLVR based methods, which is not even discussed or compared in this paper.

[1] Monte Carlo Tree Search Boosts Reasoning via Iterative Preference Learning
[2] AlphaMath Almost Zero: Process Supervision without Process
[3] Improve Mathematical Reasoning in Language Models by Automated Process Supervision

**Questions:**

Refer to weaknesses.

---

> ### Author Response · Authors · 2025-11-24
>
> **Response to W1: Novelty and Related Work.**
> We sincerely thank the reviewer for pointing out these important references [1,2,3]. These works cover preference-based policy improvement, self-evolution frameworks, and binary-search error localization, while our work targets a complementary problem: efficient generation of high-quality process supervision via adaptive resource allocation.
>
> **Fixed-budget sampling:** Prior methods allocate uniform rollouts per node (e.g., OmegaPRM [3]). Yet node uncertainty varies widely. AMCS allocates more rollouts to nodes with μ ∈ [0.4,0.6].
> We compare AMCS with OmegaPRM (OpenR implementation [1]) under different  budgets on 100 problems (200-rollout total):
>
> **Efficiency Comparison**
>
> | Metric | k=4 | k=10 | k=16 | k=22 | k=28 | AMCS |
> |--------|------|-------|-------|-------|-------|--------|
> | Avg. time (s) | 225.52 | 157.67 | 189.29 | 127.37 | 121.56 | 104 |
> | Avg. search depth | 3.72 | 3.84 | 3.48 | 3.58 | 3.78 | 3.77 |
> | Avg. explored nodes | 8 | 6.82 | 4.48 | 4.42 | 4.36 | 4.24 |
>
> Detailed analysis appears in Section 4.6.
>
> **Data Quality Analysis.** Larger k yields more accurate estimates. We compare value distributions for k=10, k=28, and AMCS:
>
> | Score Interval | k=10 | k=28 | AMCS |
> |----------------|------|-------|--------|
> | [0.0, 0.1) | 113 | 82 | 94 |
> | [0.1, 0.2) | 84 | 91 | 90 |
> | [0.2, 0.3) | 66 | 55 | 50 |
> | [0.3, 0.4) | 77 | 35 | 33 |
> | [0.4, 0.5) | 48 | 24 | 26 |
> | [0.5, 0.6) | 57 | 16 | 18 |
> | [0.6, 0.7) | 61 | 23 | 25 |
> | [0.7, 0.8) | 42 | 28 | 28 |
> | [0.8, 0.9) | 62 | 37 | 41 |
> | [0.9, 1.0] | 0 | 28 | 26 |
>
> AMCS closely matches the high-budget k=28 distribution, showing that adaptive allocation provides both efficient and accurate estimation.
>
> ---
>
> **Response to W2: Comparison with Math-Shepherd and OmegaPRM**
> We thank the reviewer for this observation.
> **Math-Shepherd** is included for reproducibility and process-data comparison.
> For **OmegaPRM**, we compare against Math-psa (OpenR [1]):
>
> | Actor Model | Strategy | Math-psa MATH500 / Omni | AMCS MATH500 / Omni |
> |-------------|----------|--------------------------|-----------------------|
> | **Phi-4-mini** | Beam | 58.5 / 13.0 | 68.5 / 16.2 |
> |  | Best-of-N | 43.7 / 13.7 | 68.0 / 16.2 |
> |  | MCTS | 65.4 / 13.0 | 69.0 / 17.2 |
> | **Llama-3.2-3B** | Beam | 62.4 / 13.5 | 61.4 / 13.7 |
> |  | Best-of-N | 55.0 / 10.5 | 59.9 / 13.2 |
> |  | MCTS | 61.4 / 11.9 | 60.0 / 12.3 |
> | **Qwen3-8B** | Beam | 50.8 / 12.3 | 49.4 / 14.1 |
> |  | Best-of-N | 46.2 / 11.8 | 47.8 / 13.9 |
> |  | MCTS | 49.4 / 13.0 | 51.2 / 14.7 |
>
> AMCS improves all OmniMATH settings and yields large MATH500 gains, particularly +24.3% with Phi-4-mini.
>
> ---
>
> **Response to W3: Experimental Results and RLVR Comparison**
> We appreciate the reviewer’s suggestion.
>
> **Performance gains.** Improvements appear across all 12 configurations, e.g., Llama-3.2-3B Beam: 54.5%→61.4%, GLM-4-9B MCTS: 70.4%→76.2%.
> **RLVR comparison.** Appendix D shows PPO with AMCS-trained PRM reaches 61.6% on MATH500 (+6.0%) and 53.1% on Hungarian Math (+6.2%). We focus on PPO due to resource limits but can extend to GRPO if required.
>
> **References:**
> [1] Wang, Y., Liu, N., Jin, Q., Zhao, W., Chen, X., & Zhang, M. (2024).  OpenR: An Open Source Framework for Advanced Reasoning with Large Language Models.  arXiv:2409.12174.
>
> [2] Lightman, H., Kosinski, M., et al. (2023).  Let's Verify Step by Step.  arXiv:2305.20050.
>
> [3] Xie, T., Chen, B., Yu, W., & Kordoni, V. (2023).  Self-Evaluation Guided Beam Search for Reasoning.  arXiv:2308.11131.

---

### Official Review · Reviewer_wmef · 2025-10-31

**Soundness:** 3
**Presentation:** 3
**Contribution:** 3
**Rating:** 4
**Confidence:** 4

**Summary:**

This paper proposes **Adaptive Monte Carlo Search (AMCS)**, a dynamic framework for generating high-quality process supervision data to train **Process Reward Models (PRMs)** for mathematical reasoning. Unlike existing methods that use fixed sampling budgets and static path expansion, AMCS introduces **uncertainty-driven adaptive sampling**—allocating more rollouts to uncertain reasoning steps—and a **temporally adaptive exploration-exploitation strategy** during path expansion, starting with broad exploration and gradually focusing on promising paths. Using AMCS, the authors construct **MathSearch-200K**, a dataset of ~200K annotated reasoning trajectories. Experiments on four mathematical benchmarks (e.g., MATH, AIME) show that a PRM trained with AMCS (**Qwen2.5-Math-7B-PRM-AMCS**) significantly outperforms existing PRMs, even enabling a 7B model to surpass a 72B model under weaker supervision. The method demonstrates strong generalization, especially on out-of-distribution and complex problems.

**Strengths:**

The paper is well-motivated, addressing a key inefficiency in the generation of data for process supervision: the uniform allocation of search budgets. The proposed AMCS method is a sound and innovative solution that dynamically allocates rollouts based on node uncertainty. In theory, this adaptive strategy should significantly improve the efficiency of data generation, allowing for more accurate value estimation and the discovery of more diverse reasoning paths within a comparable computational budget. The strong empirical results on downstream tasks suggest that the data generated by this method is indeed highly effective.

**Weaknesses:**

The primary weakness of this work lies in the experimental design, which does not strictly isolate the impact of the adaptive sampling strategy, making it difficult to draw a direct causal link between the method and the observed performance gains.

1. **Confounding Variables in Comparisons:** The main evaluation compares a PRM trained on the new AMCS-generated dataset with 200k paths against baselines trained on existing, often smaller, public datasets (e.g., 80k paths). This introduces significant confounding variables, most notably dataset size and the source of the underlying problems. Consequently, it is difficult to definitively attribute the performance improvement solely to the superiority of the adaptive sampling strategy rather than to these other factors.

2. **Lack of Robustness and Sensitivity Analysis:** The evaluation lacks a sensitivity analysis on the number of rollouts. The comparison is largely centered around a single operating point (a fixed budget of 16 rollouts), which leaves the method's performance-versus-cost trade-off unexplored. It is unclear if the reported advantages of AMCS would persist, diminish, or amplify under different computational budgets, which is a critical aspect for its practical application.

**Questions:**

1. To better isolate the effect of your adaptive sampling strategy, have you considered or performed an experiment where both the adaptive and a fixed-budget baseline generate datasets of the **same size** from the **same source problems** under a **comparable total computational budget**? Such a direct comparison would more rigorously validate the claimed benefits of adaptive sampling itself.

2. Could you provide an ablation study on the number of rollouts? A comparison between the fixed and adaptive strategies at both lower and higher rollout budgets would be highly valuable for understanding the practical applicability and robustness of your method.

---

> ### Author Response · Authors · 2025-11-24
>
> ### **Response to W1 and Q1**
>
> We appreciate the reviewer’s thoughtful analysis of our experimental design. Weakness 1 identifies confounding variables in our comparisons, while Question 1 proposes a solution through controlled experiments with identical dataset sizes and computational budgets. We address both below.
>
> **• On confounding variables (W1):**
> We acknowledge the challenge of achieving perfectly controlled comparisons. All public PRMs (Math-Shepherd, PRM800K, Deepseek-PRM) are trained on different data sizes, problem sources, and resources, making exact comparison inherently difficult without retraining all baselines (which is beyond our available resources). However, we take several measures to ensure fair evaluation:
> a. Use the same base model family as key baselines.
> b. Evaluate all PRMs on identical test sets with identical actor models.
> c. Show consistent improvements across multiple diverse baselines.
>
> **• Controlled experiment (Q1):**
> To directly address the reviewer’s concern, we conduct a controlled efficiency experiment comparing AMCS with fixed-budget OmegaPRM on the same 100 problems under the same 200-rollout budget. AMCS achieves faster generation while reaching deeper and more informative search branches, directly demonstrating adaptive allocation’s advantage independent of dataset size.  Detailed analysis and results are provided in Section 4.6 of the revised submission.
>
> ---
>
> ### **Response to W2 and Q2: Sensitivity Analysis on Rollout Budget**
>
> We thank the reviewer for this important point. Both Weakness 2 and Question 2 emphasize the need for sensitivity analysis across different computational budgets to understand AMCS’s robustness and performance–cost trade-offs. We provide a comprehensive analysis below.
>
> **Sensitivity analysis across budgets:**
> We conduct experiments comparing AMCS to fixed-budget OmegaPRM [3] (implemented using the OpenR framework [a]) with k ∈ {4, 10, 16, 22, 28} rollouts per node on 100 problems under a fixed total budget of 200 rollouts.
>
> | Budget (k) | Time (s) | Depth | Nodes |
> |------------|----------|--------|--------|
> | k = 4  | 225.52 | 3.72 | 8 |
> | k = 10 | 157.67 | 3.84 | 6.82 |
> | k = 16 | 189.29 | 3.48 | 4.48 |
> | k = 22 | 127.37 | 3.58 | 4.42 |
> | k = 28 | 121.56 | 3.78 | 4.36 |
> | AMCS | 104| 3.77 | 4.24 |
>
> ---
>
> **Reference:**
> [a] Wang et al., *OpenR: An Open Source Framework for Advanced Reasoning with Large Language Models*, 2024.

---

### Note · Authors · 2026-01-03

I have read and agree with the venue's withdrawal policy on behalf of myself and my co-authors.